# Current Considerations in Surgical Treatment for Adolescents and Young Women with Breast Cancer

**DOI:** 10.3390/healthcare10122542

**Published:** 2022-12-15

**Authors:** Brittany L. Murphy, Alicia Pereslucha, Judy C. Boughey

**Affiliations:** 1Department of Surgical Oncology, Banner MD Anderson Cancer Center, Gilbert, AZ 85234, USA; 2Department of Surgery, University of Arizona School of Medicine, Phoenix, AZ 85006, USA; 3Department of Surgery, Mayo Clinic School of Medicine, Rochester, MN 55905, USA

**Keywords:** breast cancer, adolescents and young women, surgery, outcomes

## Abstract

Adolescents and young women (AYA) with breast cancer represent a unique patient population, compared to the general population with breast cancer. We performed a literature review to evaluate the factors that influenced the surgical outcomes in this patient population. Fifty-two studies were identified, which evaluated breast surgery type, axillary surgery, contralateral prophylactic mastectomy (CPM), surgical timing, psychological factors, disparities, and imaging use. AYA patients had equivalent oncologic outcomes with breast conserving surgery (BCS) or mastectomy. CPM did not improve survival. There are limited data on axillary management in the AYA population, and while more data would be beneficial, this is currently extrapolated from the general breast cancer population. A shorter time to initiate treatment correlated to better outcomes, and disparities need to be overcome for optimal outcomes. AYA patients appreciated involvement in clinical decisions, and shared decision making should be considered whenever possible. Providers must keep these factors in mind when counseling AYA patients, regarding the surgical management of breast cancer.

## 1. Introduction

Although accounting for only ~5.6% of invasive breast cancer cases, breast cancer is the most commonly diagnosed cancer among women aged 18–39 (adolescents and young women (AYA)), accounting for 30% of cancers seen in this age group [1,2]. The risk factors for breast cancer development have been well established. The most cited factors include germline mutation status, mantle radiation prior to the age of 30, and factors that increase estrogen levels, including oral contraceptive use, early menarche, and parity after the age of 30 [3,4,5,6,7,8,9]. Other considered factors include obesity, in vitro fertilization, alcohol intake, and breast density. 

In contrast to older women, in AYA patients, obesity may be a protective factor against breast cancer as there is an association with decreased hormone synthesis [10]. There has been no proven association between breast cancer risk and in vitro fertilization [11,12,13,14]. Alcohol intake increases breast cancer risk with a 10% increase in risk per 10g of ethanol consumed per day [15]. Finally, dense breast tissue and proliferative breast disease have been found to increase breast cancer risk [16,17,18]. 

Identifying AYA patients at risk for breast cancer development is important, as 15% of deaths caused by breast cancer occur in women younger than age 45, making breast cancer the most common cause of cancer related death in this patient population [19,20,21,22]. The high mortality from breast cancer in this patient population is likely due to both the more aggressive tumor biology and because the disease is diagnosed at a later stage, due to lack of routine screening. More AYA patients are diagnosed with stage II or III disease, compared to those over age 40. Furthermore, a greater proportion of AYA patients have triple negative breast cancer or human epidermal growth factor receptor 2-positive (HER2) cancer than older women [23]. 

Previous studies have shown that a younger age is associated with a greater risk of recurrence, which is not dependent on surgery type; most of these studies, however, evaluated patients treated decades ago [24]. Furthermore, many surgical outcomes, other than oncologic outcomes, are important to consider, such as psychological factors and the impact of surgery on mental wellbeing. Therefore, we sought to perform a literature review addressing the surgical management and outcomes in AYA patients from recent publications, to gain a better understanding of the current results. 

## 2. Materials and Methods

We performed a literature review of the PubMed, Medline, and Cochrane databases, using the search terms “breast cancer” AND “adolescent young adult women” AND “treatment”, in July 2022. The search was limited from January 2012 to July 2022. This search revealed 2479 papers. Following the exclusion of non-English articles (n = 66), the remaining selection was manually reviewed by the first author, by title and abstract. The articles that did not evaluate surgical outcomes in women ≤50 years of age, were excluded (n = 2361), leaving 52 studies for review, Figure 1. The articles were classified by surgical outcome topic: breast surgery type (n = 26), axillary surgery (n = 1), contralateral prophylactic mastectomy ((CPM) n = 9), surgical timing (n = 2), psychological impact (n = 11), disparities (n = 5), and imaging (n = 1), Table 1. Of note, four studies were included in multiple categories. The major findings of these studies are discussed herein.

## 3. Results and Discussion

### 3.1. Breast Surgical Procedure

Multiple landmark studies have shown the equivalent oncologic outcomes between breast conserving surgery (BCS), with radiation and mastectomy in breast cancer, in general [76]. Most of these trials, however, included only a small proportion of young patients. In our review, twenty-six papers addressed the outcomes in the AYA population, based on the breast surgery type: BCS or mastectomy. Four studies used the Surveillance, Epidemiology, and End Results (SEER) database, with BCS rates varying between 41% and 55% [25,26,27,28]. One of these studies reported on 8656 patients, 47.7% of whom underwent BCS. A multivariable analysis revealed an improved overall survival (OS) for patients with HER2 negative disease who underwent BCS, versus mastectomy, which was thought to be related to adjuvant radiation with no significant difference seen in patients with HER2 positive disease [27]. Two of the other SEER studies also found an improved OS for BCS versus mastectomy, one of which only found this difference for patients aged 36–40, not those aged 18–35 [25,26]. The final SEER study evaluated patients with invasive lobular carcinoma, of which 41% underwent BCS and found no statistical difference in the OS outcomes by type of breast surgery [28].

Five studies evaluated data from the National Cancer Database (NCDB). One study showed that BCS was associated with an improved OS, compared to a unilateral or bilateral mastectomy, whereas two studies, one evaluating invasive disease and another ductal carcinoma in situ (DCIS) found no difference in OS, based on the surgical procedure [29,30,31]. The rates of BCS decreased over the study period in three studies, for both patients with DCIS and invasive disease [29,32,33]. The finding of more patients choosing a mastectomy over BCS in recent years was also seen in a multicenter prospective study that evaluated BCS eligible patients after neoadjuvant systemic therapy and found that the majority of patients preferred a mastectomy, independent of tumor size and regardless of treatment response [34].

Pilewskie and King performed a literature review to determine the factors that influenced the local recurrence rates in AYA patients with breast cancer. Both classic studies and modern studies were included in their review and they found that a mastectomy did not improve the recurrence rates over BCS, and furthermore, age and tumor molecular subtype were the driving forces associated with the recurrence [24]. A meta-analysis by He et al. limited to patients undergoing BCS, similarly showed age to be the biggest factor in recurrence [35]. Another meta-analysis by Chien et al. evaluated patients who underwent BCS or a mastectomy for DCIS and found lower rates of recurrence in patients who underwent a mastectomy; however, no survival benefit was found [36]. The last meta-analysis was performed by Vila et al., who evaluated outcomes in patients with invasive disease, finding no significant difference in the OS between BCS and mastectomy [37]. Similar findings were also reported in the institutional study by Cronin et al., that showed that the risk of recurrence for patients undergoing BCS for DCIS was greatest in the youngest patients [38].

Of the other evaluated studies, 10 found no difference in the OS, between BCS and mastectomy [39,40,41,42,43,44,45,46,47,49] while one noted an improved disease free survival and OS with BCS [48]. A local recurrence, however, was noted to be greater in patients undergoing BCS in two studies [44,49]. 

Overall, BCS was not inferior to a mastectomy, in AYA patients. Some studies suggested modest improved oncologic outcomes with BCS over mastectomy. There are, however, differences in the psychological effects and time to treatment of the two procedures, which will be discussed later. Patients, if candidates for either procedure, should be educated about their surgical choices, and providers should practice shared decision making. 

### 3.2. Contralateral Prophylactic Mastectomy 

It is well known that for allcomers with unilateral breast cancer, rates of CPM have increased in recent years [77]. We found nine studies that evaluated the performance of CPM in young women [50,51,52,53,54,55,56,73,74]. The rates of CPM in AYA patients have increased in recent years as well, with younger women being more likely to undergo a CPM than older women, with studies showing CPM being performed in up to 43% of AYA patients undergoing all types of surgery (BCS, unilateral mastectomy, or bilateral mastectomy) [54,56]. The factors associated with undergoing a CPM included, white race, negative lymph node status, estrogen receptor negativity, HER2 positivity, initial BCS with the need for re-operation due to positive margins, larger tumor size, lower body mass index, and testing positive for a germline pathogenic variant or having a strong family history of breast cancer [50,51,54,55,73]. Patients desired a CPM due to perceptions of decreased rates of future contralateral breast cancer and improved survival [74].

One study evaluated over 14,000 women aged ≤45 years with breast cancer who underwent a mastectomy from the NCDB, of which 29.7% underwent a CPM. Following adjustment for confounding variables, there was no difference in the OS between AYA women who underwent a CPM and those who did not [53]. Another study from a prospectively maintained institutional database showed no difference in OS between AYA patients who had a bilateral mastectomy or BCS, but patients with a unilateral mastectomy had a worse OS [52]. A single institution study from Mount Sinai Medical Center, however, showed an OS advantage with a CPM after 10 years of follow up [56].

Overall, there was no dominant clear evidence that a CPM improved survival in the AYA patient population. Providers should ensure that this is clear to patients, especially if impact on survival is one of the patient’s motivations for choosing a CPM. 

### 3.3. Axillary Surgery

The use of sentinel lymph node (SLN) surgery, to evaluate nodal disease in clinically node negative patients with breast cancer, has been well established. No studies were identified that focused on SLN use in the AYA population; therefore, management is the same as it is for the general population of patients with breast cancer. Previous studies that evaluated axillary nodal management in clinically node negative patients with a small volume nodal disease at the time of upfront surgery included only a small proportion of AYA patients. The youngest patient in the ACOSOG Z0011 trial was 24, but only approximately 35% of patients were under 50 years of age [78]. In the AMAROS trial, the youngest patient was 48 years old [79]. Prememopausal patients composed 31% of the patient population in the OTOASOR trial, with the youngest patient being 26 years of age [80]. In our review, only one study addressed management of the axilla for young women clinically node negative at the time of diagnosis, with one to three positive SLNs, at the time of upfront surgery. Of the 357 women included in the study, 192 underwent BCS and 165 underwent a mastectomy. Use of the completion axillary lymph node dissection (ALND) was significantly higher in patients who underwent a mastectomy (144/165, 87%), than in patients who underwent BCS (10/192, 5.2%). Regional nodal irradiation was given to a greater proportion of patients who underwent a mastectomy than BCS (48% vs. 30%, *p* < 0.01) [57]. This reflects the adoption of the ACOSOG Z0011 findings into the AYA population and follows a similar trend to the general population with breast cancer [81].

These findings and the paucity of data on all aspects of axillary management in the AYA population, highlights the need for continued research, as it is possible that patients who undergo both an ALND and radiation for small volume axillary disease are being over-treated, which may contribute to additional morbidity for a limited oncologic benefit. Omission of completion ALND following a mastectomy, for a small volume nodal disease in patients who will be receiving regional nodal radiation, is slowly being accepted as standard treatment for allcomers with breast cancer [81].

### 3.4. Time to Treatment 

Two studies addressed time to initiation of treatment. One study determined that the time to initiate treatment was longer by one week with upfront surgery, compared to the time to the initiation of neoadjuvant chemotherapy. The delay was attributed to fertility preservation, as a greater proportion of patients in the neoadjuvant chemotherapy group declined fertility preservation, although this could also be related to access to plastic surgery or access to the operating room [58]. Another study evaluated the California Cancer Registry database and determined that for patients treated with upfront surgery, a time to treatment of greater than 6 weeks was associated with an 80% 5-year survival versus a 90% 5-year survival when the time to surgery was less than 2 weeks. A greater proportion of patients undergoing BCS or a modified radical mastectomy were able to enter the operating room within 2 weeks, than a total mastectomy; reconstruction was not discussed, which can often contribute to a longer time to surgery [75].

These studies highlight that coordination of care to allow for efficient initiation of treatment is of upmost importance to AYA patients. AYA patients should be evaluated by the multidisciplinary team to determine the order of treatment, to arrange for genetic testing, and to evaluate for fertility preservation in a timely manner as to initiate treatment as soon as possible. 

### 3.5. Psychological Effects of Surgery

Surgery creates permanent changes to a patient’s body. Eleven studies evaluated the psychological effects of breast cancer surgery on AYA patients. Multiple studies showed that a mastectomy was associated with worse quality of life factors, such as body image, sexual health, and anxiety, compared to patients who underwent BCS [59,60,61]. Many patients chose mastectomy out of fear of potentially leaving cancer cells behind and the desire to avoid future screening imaging [62]. The increased piece of mind gained from a mastectomy was similar to the reasons for why patients chose to undergo a CPM [62,63,74]. Patients who underwent a mastectomy with reconstruction had better quality of life scores with less stress and anxiety, but endorsed more physical discomfort in the post operative setting [64]. The barriers to reconstruction following a mastectomy included the fear of cancer relapse and financial concerns [65]. The physical side effects of radiation following BCS was also a concern, but the quality of life scores were shown to return to the reference ranges three years following the completion of radiation [66]. 

Other studies evaluated the patient role in decision making and patient education. Patients in the AYA population were found to appreciate more information about treatment options and anticipated outcomes, with worse reported quality of life scores when treatment decisions were made for them, rather than with them [67,68]. This finding again highlights the importance of shared decision making in the AYA patient population.

### 3.6. Disparities

Acknowledging the disparities in healthcare is important, in order to overcome the limitations. Five studies evaluated the disparities in AYA patients who underwent surgery for breast cancer [69,70,71,73,75]. As previously discussed, the timing to surgery influenced the oncologic outcomes in one study. That same study found that Hispanic and African American patients (compared to non-Hispanic white), patients with public or no insurance (compared to private insurance), and patients with a low socioeconomic status (compared to a high socioeconomic status) were more likely to have a treatment delay of greater than 6 weeks, which was associated with worse outcomes [75]. In other studies, surgical management was found to be equivalent in public and private hospitals [69]; however, race was associated with undergoing a CPM, with white women being twice as likely to undergo a CPM than women of other racial groups [73]. The oncologic outcomes were found to vary with marital status, with a survival benefit of BCS over mastectomy observed in unmarried patients [71].

The financial burden of healthcare is another important factor to consider, especially as financial concerns influence a patient’s surgical reconstruction choices, as previously discussed. A study that explored the financial toxicity determined that a younger age, non-white race, and a lower socioeconomic status, was associated with a higher financial toxicity, regardless of the actual out of pocket costs [70].

These factors must be considered when treating AYA patients, as we need to ensure equal high-quality care to all, with as minimal financial burden as possible. Providers should ensure access to social services, to help provide resources to interested patients. 

### 3.7. Imaging

It is well known that breast cancer detection is lower in women with dense breast tissue, and dense breast tissue is associated with younger age. The COMICE trial was a randomized controlled trial that determined that a pre-surgical MRI did not reduce reoperation rates for patients undergoing BCS; however, only 23% of their patient population was under 50 years of age [82]. Park et al. evaluated patients 35 years of age or younger, who underwent treatment at a single institution, and showed that a MRI identified additional sites of disease in 13% of patients, but it did not increase mastectomy rates, and reduced reoperation rates, suggesting that MRIs should be considered in young women undergoing BCS [72].

## 4. Conclusions

AYA patients who are candidates for either BCS or mastectomy may undergo either procedure with equivalent outcomes. The outcomes with BCS may be optimized with a pre-operative MRI. A CPM did not improve survival and more studies are needed to determine optimal management of the axilla in the setting of low volume nodal disease at upfront surgery. Providers must keep in mind that a shorter time to initiate treatment correlates to better outcomes and disparities need to be overcome. AYA patients appreciate involvement in clinical decisions and shared decision making and should be considered whenever possible. Providers must keep these factors in mind when counseling AYA patients regarding the surgical management of breast cancer.

## Figures and Tables

**Figure 1 healthcare-10-02542-f001:**
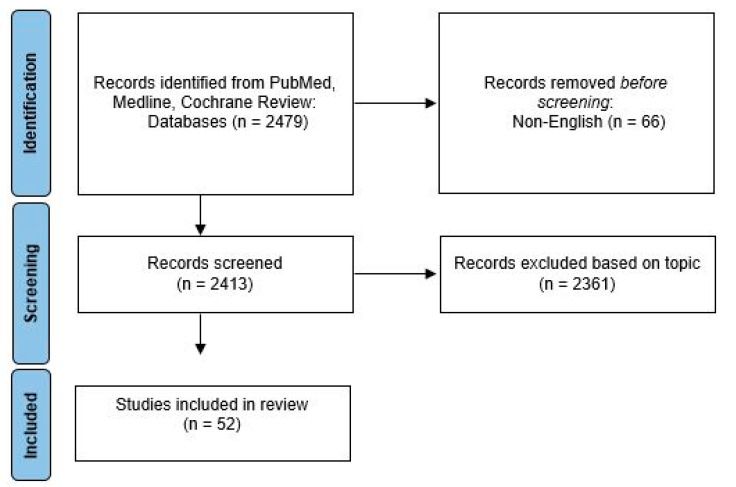
Flow diagram of the included studies.

**Table 1 healthcare-10-02542-t001:** Studies included in the literature review.

Authors	Year	Journal	Outcome Evaluated
Pilewskie M, King TA [24]	2014	J Surg Oncol	Surgery-Breast
Sun ZH, Chen C et al. [25]	2021	Medicine	Surgery-Breast
Ye JC, Yan W et al. [26]	2015	Clin Breast Cancer	Surgery-Breast
Yu P, Tang H, et al. [27]	2020	Cancer Control	Surgery-Breast
Yu TJ, Liu YY et al. [28]	2018	Eur J Surg Oncol	Surgery-Breast
Byun DJ, Wu SP, et al. [29]	2021	Ann Surg Oncol	Surgery-Breast
Lazow SP, Riba L, et al. [30]	2019	Breast J	Surgery-Breast
Orozco JI, Keller JK et al. [31]	2022	Ann Surg Oncol	Surgery-Breast
Pesce CE, Liederbach E et al. [32]	2014	J Am Coll Surg	Surgery-Breast
Rutter CE, Park HS et al. [33]	2014	Ann Surg Oncol	Surgery-Breast
Kim HJ, Dominici L et al. [34]	2022	Ann Surg	Surgery-Breast
He XM, Zou DH [35]	2017	Sci Rep	Surgery-Breast
Chien JC, Liu WS et al. [36]	2022	Breast	Surgery-Breast
Vila J, Gandini S et al. [37]	2015	Breast	Surgery-Breast
Cronin PA, Olcese C et al. [38]	2015	Ann Surg Oncol	Surgery-Breast
Bao S, He G [39]	2022	Breast J	Surgery-Breast
Cao JQ, Truong PT et al. [40]	2014	Int J Radiat Oncol	Surgery-Breast
Chen LJ, Chang YJ et al. [41]	2021	BJS Open	Surgery-Breast
Frandsen J, Ly D et al. [42]	2015	Int J Radiat Oncol Biol Phys	Surgery-Breast
Jeon YW, Choi JE et al. [43]	2013	Breast Cancer Res Treat	Surgery-Breast
Maishman T, Cutress RI et al. [44]	2017	Ann Surg	Surgery-Breast
Quan ML, Paszat LF et al. [45]	2017	J Surg Oncol	Surgery-Breast
Sinnaduri S, Kwong A, et al. [46]	2019	BJS Open	Surgery-Breast
Wang L, He Y et al. [47]	2020	Breast Cancer Res Treat	Surgery-Breast
Li P, Li L et al. [48]	2022	Front Oncol	Surgery-Breast
Xie Z, Wang X et al. [49]	2014	Ann Surg Oncol	Surgery-Breast
Bouchard-Fortier A, Baxter NN et al. [50]	2018	Curr Oncol	Contralateral Prophylactic Mastectomy
Dettwyler SA, Thull DL et al. [51]	2022	Breast Cancer Res Treat	Contralateral Prophylactic Mastectomy
Donovan CA, Bao J et al. [52]	2017	Ann Surg Oncol	Contralateral Prophylactic Mastectomy
Pesce CE, Liederbach E et al. [53]	2014	Ann Surg Oncol	Contralateral Prophylactic Mastectomy
Rosenberg SM, Sepucha K et al. [54]	2015	Ann Surg Oncol	Contralateral Prophylactic Mastectomy
Terkelsen T, Ronning H, et al. [55]	2020	Acta Oncol	Contralateral Prophylactic Mastectomy
Zeichner SB, Ruiz AL et al. [56]	2014	Asian Pac J Cancer Prev	Contralateral Prophylactic Mastectomy
Tadros AB, Moo TA et al. [57]	2020	Breast Cancer Res Treat	Surgery-Axilla
Guay E, Cordeiro E et al. [58]	2022	Ann Surg Oncol	Surgical Timing
Berhili S, Ouabdelmoumen A et al. [59]	2019	Clin Breast Cancer	Psychological
Olasehinde O, Arije O et al. [60]	2019	J Glob Oncol	Psychological
Rosenberg SM, Dominici LS et al. [61]	2020	JAMA Surg	Psychological
Rosenberg SM, Greaney ML et al. [62]	2019	J Adolesc Young Adult Oncol	Psychological
Rosenberg SM, Greaney ML et al. [63]	2018	Psychooncology	Psychological
Fanakidou, I, Zyga S, et al. [64]	2018	Qual Life Res	Psychological
Nozawa K, Ichimura M et al. [65]	2015	Int J Clin Oncol	Psychological
Bantema-Joppe EJ, de Bock GH et al. [66]	2015	Br J Cancer	Psychological
Recio-Saucedo A, Gerty S et al. [67]	2016	Breast	Psychological
Seror V, Cortaredona S, et al. [68]	2013	Psychooncology	Psychological
Patel A, Wang WJ et al. [69]	2019	Breast J	Disparity
Politi MC, Yen RW et al. [70]	2021	Oncologist	Disparity
Zhang J, Yang C et al. [71]	2021	Front Oncol	Disparity
Park AR, Chae EY et al. [72]	2021	Radiology	MRI Evaluation on Breast Surgery
Grimmer L, Liederbach E et al. [73]	2015	J Am Coll Surg	Multiple-Contralateral Prophylactic Mastectomy and Disparity
Rosenberg SM, Tracy MS et al. [74]	2013	Ann Intern Med	Multiple-Contralateral Prophylactic Mastectomy and Psychological
Smith EC, Ziogas A et al. [75]	2013	JAMA Surg	Multiple-Surgical Timing and Disparity

## Data Availability

Not applicable.

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
