# Peer review of "Current Considerations in Surgical Treatment for Adolescents and Young Women with Breast Cancer"

_healthcare, 2022, doi:10.3390/healthcare10122542_

Round 1
Reviewer 1 Report
Most of our knowledge about breast cancer is based upon studies in older women, while young women are under-represented in contemporary research. Trials dedicated to young patients are sheerly needed to answer many clinical questions and to ensure appropriately tailored treatment, particularly to avoid over-treatment based exclusively on age considerations.
Management of young women with breast cancer is a very complex task, that requires a dedicated care, which is only possible in a multidisciplinary setting, where health professionals have been exposed to specific training and clinical experience.
Young women with BC represent a very particular group of patients: personalized psychosocial support, counselling on genetic predisposition, fertility, sexual health, and socio-economic consequences must be incorporated into individual treatment planning, while specific guidelines for post-treatment survivorship must be considered when caring about young survivors.
For specialists, but also for primary care clinicians, in order to better manage young BC patients and survivors, comprehensive management strategy and guidelines must be available to ensure optimal care and outcomes.
This metanalysis fits perfectly in this this context and adds knowledge to a field which is still far from being adequately addressed. The work is very well structured and, in my opinion, can be considered for publication as it is.
Author Response
Thank you for your review and kind words.
Reviewer 2 Report
Dear Authors,
I congratulate to authors on this comprehensive review. I have only two comments.
1. You stated that mastectomy and BCS have equivalent outcomes, however certain parameters (e.g. psychological consequences and time to treatment) may differ. If you think the two procedures are not fully equivalent, I suggest you add a comment to that effect. It can be important in the joint decision-making process.
2. I advise you to add the results of the OTOASOR study (DOI:10.1016/j.ejso.2016.12.011) to the armpit surgery section. Although it was only performed in one center in Hungary and the number of participants was smaller than in the other two studies (244+230 patients), the youngest patients were 26 and 27 years old, and the proportion of premenopausal patients was 34% and 27% in the two arms, respectively.
Author Response
We appreciate the reviewer's time and feedback. The recommended changes were made to the manuscript, as discussed below.
- You stated that mastectomy and BCS have equivalent outcomes, however certain parameters (e.g. psychological consequences and time to treatment) may differ. If you think the two procedures are not fully equivalent, I suggest you add a comment to that effect. It can be important in the joint decision-making process.
Thank you for this observation. The surgery section was updated to include "oncologic" prior to outcomes to clarify. Furthermore, an introduction to the discussion of psychological and time to treatment outcomes was added to the surgery section.
2. I advise you to add the results of the OTOASOR study (DOI:10.1016/j.ejso.2016.12.011) to the armpit surgery section. Although it was only performed in one center in Hungary and the number of participants was smaller than in the other two studies (244+230 patients), the youngest patients were 26 and 27 years old, and the proportion of premenopausal patients was 34% and 27% in the two arms, respectively.
Thank you for this recommendation. This study was added to the axillary surgery section, in a similar manner as the Z11 and AMAROS trials.